Can head louse repellents really work? Field studies of piperonal 2% spray

Burgess Ian F. 1 ian@insectresearch.com
Brown Christine M. 1
Burgess Nazma A. 1
Kaufman Judith 2
1 Medical Entomology Centre, Insect Research & Development Limited , Quy Court, Stow-cum-Quy, Cambridge , UK
2 Royalheath Charitable Trust Limited , London , UK
Andrew Nigel
Electronic publication date: 2014 Apr 17
Publication date: 2014
Volume: 2
Electronic Location ID: e351
Received 2014 Jan 17; Accepted 2014 Mar 28
Copyright: © 2014 Burgess et al.
Copyright year: 2014
Copyright holder: Burgess et al.
License: This is an open access article distributed under the terms of the Creative Commons Attribution License, which permits unrestricted use, distribution, and reproduction in any medium, provided the original author and source are credited.
License URL: https://creativecommons.org/licenses/by/3.0/

Keywords: Repellent, Head lice, Piperonal, Field study, Consumer use

Funding: Charwell Pharmaceuticals Ltd The initial laboratory investigations and field studies were supported financially by Charwell Pharmaceuticals Ltd, Alton, UK, as part of a new product development programme prior to their acquisition by Pfizer Inc., and inclusion in Unicliffe Limited. Employees of Charwell Pharmaceuticals Ltd filed patents based on the work in support of a commercial product launch, but that intellectual property has now lapsed. The company did contribute ideas towards the design of the studies and conducted their own analyses of the results, which were not disclosed to the investigators. David Merrington contracted the work and William (Bill) Oliver and Richard Irwin provided technical and logistical support and study supplies. Monitoring of documentation from both field studies was performed by an appointee of Pfizer Inc. on behalf of Charwell Pharmaceuticals Ltd.

==============================
Background. Many families find regular checking of children’s heads for head louse infestation too onerous and would prefer to be able to prevent infestation by use of a topical application that deters lice from infesting the head. Identification in the laboratory of a repellent activity for piperonal provided the basis for developing a spray product to repel lice.

Methods. A proof of principle field study in Dhaka, Bangladesh, compared the effect of using 2% piperonal spray with that of a placebo in 105 children and adults from three communities with infestation levels close to 100%. All participants were treated for infestation and subsequent incidence of reinfestation monitored daily by investigators. A second randomised, controlled, double blind, study in North London, UK, evaluated the effect of the product in normal use. One hundred and sixty-three children from schools with a high level (20–25%) of infestation were treated and confirmed louse free and randomly divided between 2% piperonal, a placebo spray, and a control group for up to 22 weeks. Parents applied the spray and monitored for infestation. Regular investigator visits confirmed the parental monitoring and replenished supplies of spray.

Results. In Dhaka, over 18 days there were only 4 infestations in the piperonal group and 8 in the placebo group. This difference was not significant (p = 0.312). In North London, there were 41 cases of infestation over the course of the study. Although there were fewer infestations in the piperonal group, analysis of time to first infestation showed a no significant (p = 0.4368) difference between groups.

Conclusion. Routine use of 2% piperonal spray in communities with a high prevalence of head louse infestation may provide some protection from infestation. However, the difference between use of the product and no active intervention was sufficiently small that regular checking for presence of lice is likely to be a more practical and cost effective approach to prevention of infestation.

Introduction

Most human management of head lice involves treatment post-infestation, either by combing or other physical removal or using various types of insecticidal chemicals. Successful interventions often depend upon timely diagnosis of infestation before it becomes established. Over the years, health educators have encouraged regular and frequent checking of children’s hair for signs of infestation but with limited success because people are either too busy or not concerned enough about lice. They would rather deal with the problem if and when it arises.

Most parents would like a way to prevent lice from infesting the hair. The majority ideal is a product that stops lice transferring from one host to another. Of course, materials can be applied to make the hair unacceptable as a habitat but they are also mostly unacceptable for cosmetic reasons, such as heavy vegetable oils like coconut, neem, olive, and sassafras oils, that attract dirt, render hair lank and greasy, and develop distasteful odours after a short time on the head.

The idea of a louse repellent was quite novel when this investigation started in 1989 (Burgess, 1993a), although suggestions that some essential oils had repellent properties had circulated for years before (Spencer, 1941). At the time the idea was sufficiently novel that the concept needed careful explanation to health care professionals. Previously only residual insecticides were thought to confer some measure of protection against reinfestation (Burgess, 1993b; Peock & Maunder, 1993). While investigating discontinued pediculicides we found that 1,3-benzodioxole-5-carbaldehyde (piperonal or heliotropin), a fragrance and flavouring agent with an odour similar to vanilla, deterred lice from walking onto surfaces treated with it (Burgess, 1993a). An extensive investigation of this and related chemicals led to development of a repellent product (Irwin, 1992; Irwin, 1993; Oliver, 1992; Peock & Maunder, 1993).

A 2% piperonal spray was marketed in Britain from late 1992 but, as a head louse control product its status was questioned because it did not have a Marketing Authorisation from the Medicines Control Agency (MCA), even though no claims of pediculicidal activity were being made. The MCA initially stated that as a repellent the product was not licensable (no mosquito repellents were licensed at the time) but they reserved the right to change this viewpoint so the manufacturer prepared a pharmaceutical dossier should it be required, which necessitated preparation of a clinical evaluation report.

This report describes two studies, one in Bangladesh, and the other in the UK. The main objective for both studies was to determine whether 2% piperonal spray could protect against contracting head louse infestation, with the expectation that regular use could reduce the risk of becoming infested. An additional aspect for the UK study was to determine whether use of the repellent on a regular basis was a practical proposition for parents and guardians when they were busy preparing their children for school each day.

Materials and Methods

We conducted two field studies. The first was in Dhaka, Bangladesh, where reinfestation risk was high. The second in North London, UK, enabled us to evaluate effectiveness over time. Anyone wishing to take part that was found to be infested at the start of the study was treated so that all participants started louse free.

Settings and participant flow

Study 1: In Dhaka, between 4th February and 10th of March 1993 we recruited 107 participants from three communities where, from previous experience, we knew infestation was close to 100% prevalence. These were two religious-based orphanages at Farmgate and Mohammepur in Dhaka city and a 7000 population bostee (slum) community at Gandaria, between Dhaka and Naryanganj. An information leaflet was translated into Bengali by one of us (NAB) and distributed through the institution administrators and the community chairmen. Verbal explanation of the study requirements was provided for anyone unable to read.

Pre-enrolment screening used a plastic detection comb method that has since been shown to be 3.84 times more effective than visual inspection (Balcioglu et al., 2008). Of the 92 residents in the Farmgate orphanage we screened 70 children and found 68 to be infested. The remainder declined examination. From these we recruited 6 males and 38 females. In Mohammedpur, 160 children were registered. We examined 80 using the same method, all had lice, and 21 agreed to enrol in the study. Those not screened were not in the building at the time. Here we recruited 9 males and 10 females, with ages ranging from 7 to 16 years, with one adult participant. Hair length was long for 41/48 females (85.4%), with three having medium and the remainder short hair. All males had short hair.

Participants were allocated to receive either the 2% piperonal spray or the placebo using the anonymously labelled, randomised bottles supplied by the sponsor. Because fewer than half the residents at each site agreed to participate in the study no more than 50% of children sharing a dormitory were participants, which we considered adequate to allow opportunities for reinfestation.

Because administrative problems had delayed regulatory release of the study materials, it became necessary to shorten the time allocated to each treatment phase from the planned 14 days to nine. At day 10, during the cross-over, we found minimal louse transmission had occurred in the children’s homes so the Gandaria site was initiated to provide additional data. Everybody examined at this site was found to be infested. To increase the risk of reinfestation we recruited only one person from each household. This site operated for 9 days, in parallel with the second half of the cross-over in the orphanages. At Gandaria, all 42 participants were female aged from 7 upwards, with 14 adult participants

At all three sites, continuity was disrupted by participants ending participation or visiting their extended families for the month of Ramadan. Even Christian children took time off to visit family members. Consequently, analyses were conducted on the ITT population only.

Study 2: We had previously worked with the Orthodox Jewish community in the Golders Green, Hendon, and Edgware districts of North London, such as in the first identification in the UK of acquired resistance to pyrethroid insecticides in head lice (Burgess et al., 1995). Prevalence of infestation in one school averaged 20% to 25% and many families in the community expressed the belief at public meetings that treating children was pointless because reinfestation occurred within days.

Most participants attended a primary school that distributed an invitation letter, study information, and a Consent form, which had previously been discussed with members of the community. Others heard about the study from friends and neighbours. All were pre-assessed for suitability by their general practitioner.

At this site recruitment was based around the family, with 163 children from 48 families taking part. Households ranged in size from three to 17 members, the most common being 8 people (11 households), followed by five households each for 7, 9, 11, and 12 members, four households with 6 people, three each for 5 and 10 people, two each for 4 and 13 members, and one household each for 3, 15, and 17 members. Numbers of participants per household ranged from one to seven with 13 households having 3 participants and 11 having 4, there were eight families each with 2 and 5 participants, five with 1, 2 with 6, and only one with 7 taking part.

The population comprised 112 (68.7%) females and 51 males. All participating boys had short hair and among the girls 34 (30.4%) had long hair, 70 (62.5%) had medium length, and just eight (4.9%) had short hair.

At this site we planned the study for between 6 and 13 weeks, although the protocol allowed this period to be extended. It actually ran over 22 weeks, between 29th May and 11th November 1994. Using a rolling enrolment, the initial distribution of participants was 53 allocated 2% piperonal spray, 48 allocated placebo spray, and 43 in the control group. Over the full study period we recorded 41 infestations for the time-to-first-infestation analysis. Some of the participants who caught lice opted to continue in the study in a different randomisation group but not all volunteers actively participated for the whole time and the intention to treat population included families who dropped out for various reasons during the summer months. Some procedures were disrupted by religious festivals during the study period. We did not analyse the outcomes in those reallocated to the alternative study groups because too few people chose to remain in the study to permit meaningful analyses to be carried out, especially since the majority of them were in the no intervention monitoring group.

All participants gave baseline data on age, gender, hair characteristics. In Dhaka, participants were photographed and also gave their father’s or husband’s name (a local cultural practice) for later confirmation of their identity in the large communities where family names are rarely used.

The lower age limit was 4 years, with an upper limit of 14 years in London but there was no upper limit in Dhaka. All treatments and assessments were domiciliary, except at Gandaria where we used a community clinic to examine and treat participants.

Inclusion criteria were fitting the age profile; normal physical health; willingness to participate and to be treated for lice. Exclusion criteria were a history of allergy, asthma, eczema, contact dermatitis or psoriasis; or concomitant steroid use. Participation in North London was subject to GP approval.

Ethics

Ethical approval in Dhaka was granted by the ad hoc ethics committee of the Metropolitan Medical Centre, Mohakhali, Dhaka; Protocol RAP001. Study medications were granted access to the country by the Directorate of Drug Administration of the Ministry of Health and Family Welfare, Bangladesh.

Consent to treat and participate was provided en bloc in the two orphanages by administrators, acting in loco parentis. Also each volunteer was counselled and gave a witnessed signed assent to participate. In Gandaria, participants provided a signed/marked assent prior to enrolment.

Ethics approval in North London was granted by Barnet Research Ethics Committee; Protocol RAP002. A Clinical Trial Exemption Certificate (CTX) was granted by the MCA. Parents provided written consent for all children of their household. Anyone unwilling to participate and ineligible household members could join a monitoring group to provide information on the background infestation risk in the community.

The studies were conducted in conformity with the principles of the Declaration of Helsinki and the OECD Guidelines for Good Clinical Practice (GCP) prevailing at the time, which are no longer available but which were embodied in the International Conference on Harmonization guideline on Good Clinical Practice E6(R1) (ICH-GCP, 1996).

Study medications

The investigational spray was a marketed general sales list (GSL) product containing 2% piperonal in an aqueous alcohol base (Rappell®, Charwell Pharmaceuticals Ltd, UK). It was supplied in 90 ml pump spray plastic bottles delivering metered 130 µl doses. The recommended application rate stated by the manufacturer on the product label was 5–25 sprays daily, according to the length and thickness of hair, before school or other activities. So for a boy with a 5 mm long cropped hair the minimal dose would be applied whereas for a girl with thick hair that hung below the shoulders the maximum application would be necessary. The product could be reapplied if the hair was wetted during the day, e.g., after swimming.

The placebo comparator was a superficially identical spray containing 1% vanillin to mimic the odour. Previous laboratory tests had found vanillin was not repellent (Irwin, 1992; Irwin, 1993; Oliver, 1992).

We used carbaril 1% aqueous emulsion (Derbac-C liquid; Charwell Pharmaceuticals Ltd), which was found in a series of comparative tests to be effective with a single application and left no insecticide residue (Burgess, 1990), to eliminate lice before using the investigation products or if lice were caught during the study.

In Bangladesh all medications were applied by investigators. After treatment, participants were checked to confirm efficacy and the allocated spray applied daily by an investigator. Assessments were made on alternate days using visual inspection and detection combing.

In North London the sprays were applied by a parent. A louse detection comb was supplied so they could check for lice, at least three times weekly. The parent noted on a diary card when spray was applied and when they checked for lice. If lice were found our pharmacist investigator (JK) supplied insecticide treatment. An investigator visited every family once each month to make an independent check for lice, collect diary cards, and replenish the spray.

Outcomes

The primary outcome measure was the time to first infestation with head lice, confirmed by detection combing. Secondary endpoints were whether infestations occurred at any time while using the product, and the safety of the spray in use.

Sample size

There were no precedents for estimating sample size. No studies at that time had ever been conducted on incidence of head louse infestation in any community, and assumptions of risk had never been quantified. Consequently, we assumed that in populations with a high prevalence of infestation there would be sufficient reinfestation risk that protective activity would be detectable in a relatively small population. That assumption has since been partially confirmed by a recent study conducted in Brazil showing that, in a high prevalence population, reinfestation is likely to occur in around 14–24 days (Pilger et al., 2010).

The Dhaka study was a proof of concept comparing the active spray with placebo, with underlying prevalence close to 100% in participating communities, as determined by scalp examination and detection combing. We estimated that recruiting up to half the children in the orphanages would provide a reasonable risk of reinfestation from other residents. In Gandaria, we recruited a cohort equal to the larger orphanage group.

In North London the protocol provided for recruitment of up to 100 participants per treatment or control group (i.e., up to 300 participants in total). It was not clear whether a sample size estimation was conducted on behalf of the sponsor because no details were included in the protocol and no specific information was conveyed either formally or informally to investigators.

Randomisation—allocation concealment

The proof of concept employed a randomisation sequence in which treatment allocation was predetermined and concealed, with bottles anonymously labelled “A” or “B”. In the orphanages, each participant acted as their own control using a cross-over to the other spray half-way through the allotted period.

We planned each treatment phase for 14 days, which was reduced to nine days for logistical reasons. In Gandaria a cross-over was not practicable so treatments were allocated by pairs of individuals. As everyone lived in similar circumstances we considered that risk factors for infestation were essentially similar for all participants, thereby “matching” the individuals in the pairs.

Randomisation in North London was by family, using a computer generated allocation sequence composed of balanced blocks of eight, i.e., each household constituted one block. Treatments were labelled with coded identification numbers, so investigators and participants were both blind to the allocation.

This study operated a form of cross-over design but at this site each participant used the same preparation until they became infested or reached the end of the study period. Participants in either spray group who became infested could cross-over to the non-intervention group (Fig. 1). Participants in the non-intervention group who became infested could cross-over to a randomised spray group provided they were eligible.

Figure 1 Flowchart of participants in the London study.

Product codes were not broken until after completion of data collection, entry into the study database, and database lock.

Statistical analysis

The protocol stated that BIOS (Consultancy & Contract Research) Ltd were to analyse data and prepare a report on behalf of the sponsor, Charwell Pharmaceuticals Ltd. However, as far as we are aware, no formal statistical report was produced by that consultant and no report of any kind was made available to the investigators by the sponsor.

We conducted a post hoc analysis for the primary outcome in which Kaplan–Meier curves have been used to illustrate the time pattern of participants remaining free from infestation when using either 2% piperonal or placebo sprays, or where no intervention was used.

We conducted analyses based on both the intention-to-treat (ITT) and the per-protocol (PP) populations (Altman, 1991; Kirkwood & Sterne, 2003). Analysis of data from Dhaka took into account the majority cross-over design so we tested binary outcomes using the McNemar test and, due to the low number of events, essentially evaluated whether an infestation occurred at all (Klingenberg & Agresti, 2006; Klingenberg et al., 2009). Analyses of counts or ranked data used the Wilcoxon signed rank test for paired data. However, because the three curves from North London data were independent, being based on different participants, it was possible to use the log-rank test to test differences between treatments for significance (Bland & Altman, 1998). There were insufficient data available to conduct demographic analyses.

Results

Outcomes

Dhaka

One reason these communities were selected for the study was that, given the high prevalence of head louse infestation in each community, we anticipated that those we treated ran a high risk of catching lice from their untreated peer group. Consequently, we expected an incidence of several cases each day, especially in the placebo treated group. However, the rate of reinfestation observed at all three sites was surprisingly low compared with expectation, particularly in the orphanages where nobody slept in individual beds and children routinely gathered to watch us with their heads together. We saw similar clusters of curious onlookers at Gandaria, where family members slept in close proximity in each household.

Only 12 reinfestation events occurred. One boy caught lice on both phases of the cross-over and one pair (one active and one placebo) of the parallel group participants also caught lice. Four participants were infested using placebo but not using piperonal and two from the parallel group were infested using placebo. Two were infested using piperonal but not using placebo. This gave 8 infestations using placebo and 4 using active. Comparison of the Kaplan–Meyer curves (Fig. 2) for protection against infestation using a log-rank test showed a non-significant difference (chi-squared = 1.577, p = 0.2091) between the piperonal and placebo sprays, although in part these data were not strictly independent. If compared by Wilcoxon signed rank analysis the outcome was also non-significant (z = −1.0097, p = 0.312). The application rate for the spray averaged 2.37 g daily per participant.

Figure 2 Kaplan–Meyer plot of the proportion of participants louse free in the Dhaka study.

North London

In order to show parents that reinfestation did not occur as rapidly as believed, we set up a small programme for 22 children from 10 closely associated families to monitor incidence of infestation. Anyone with lice was treated to eliminate infestation and then confirmed to be louse free. The parent then checked the children using the detection comb at least once weekly. If lice were found they were treated after which the monitoring continued. All the children were followed over 9 weeks, showing that reinfestation was considerably less likely than anticipated, with no infestations until the third week. Overall there were 13 cases of infestation in nine individuals, with four children from two families being infested twice (Table 1). These data suggested that transmission within households was more common but we were unable to identify links to explain the importation of lice into any of the households.

Table 1 Outcomes from the preliminary investigation to monitor reinfestation rates in the North London community.

Participant	Infestation found	
Family	Child	At start	1	2	3	4	5	6	7	8	9	
A	1	No	-	-	-	-	-	-	-	Yes	-	
	2	No	-	-	-	-	-	-	-	-	-	
B	1	No	-	-	-	-	-	-	Yes	-	-	
C	1	No	-	-	-	-	-	-	-	-	-	
	2	No	-	-	-	-	-	-	-	-	Yes	
D	1	No	-	-	-	-	-	-	-	-	Yes	
E	1	No	-	-	-	-	-	-	-	-	-	
F	1	No	-	-	Yes	-	-	-	-	Yes	-	
	2	No	-	-	-	Yes	-	-	-	Yes	-	
	3	No	-	-	Yes	-	-	-	-	Yes	-	
G	1	No	-	-	-	-	-	-	-	-	-	
	2	No	-	-	-	-	-	-	-	-	-	
	3	No	-	-	-	-	-	-	-	-	-	
	4	No	-	-	-	-	-	-	-	-	-	
H	1	No	-	-	-	-	-	-	-	-	-	
	2	No	-	-	-	-	-	-	-	-	-	
	3	No	-	-	-	-	-	-	-	-	-	
J	1	No	-	-	-	-	-	-	-	-	-	
	2	No	-	-	-	-	-	-	-	-	-	
	3	No	-	-	Yes	-	Yes	-	-	-	-	
K	1	Yes	-	-	-	-	-	-	-	-	Yes	
	2	Yes	-	-	-	-	-	-	-	-	-	
Weekly incidence %			0	0	13.6	4.5	4.5	0	4.5	13.6	13.6	
Cumulative incidence%			0	0	13.6	18.2	22.7	22.7	27.2	45.5	59.1	

Intention to treat comparison of the three treatment groups for time to first infestation by log-rank analysis showed fewer participants caught lice when using repellent but this was not significant (chi-squared = 1.6567, p = 0.4368). Figure 3 shows the Kaplan–Meyer curves of probability of remaining louse free for the ITT group over the 22 weeks. The PP analysis was essentially similar with no significant difference between the groups (chi-squared = 2.2035, p = 0.3323).

Figure 3 Kaplan–Meyer plot showing the proportion of participants remaining louse free in the London study.

Some families applied the spray conscientiously throughout the study period. Others found the need for daily application too burdensome in a busy household with numerous children. Consequently, spray use was inconsistent, although residues of the piperonal could persist for a few days since most children on the study washed their hair just once each week. We observed that fine hair looked greasier than normal, which was resolved by reducing the application rate. We could not estimate the daily application rate due to inconsistencies of use and because some of the bottle weight data were not returned to the investigation site by the sponsor. However, the sponsor reported that most parents applied less spray than they thought they had, and this was apparent from the partial data available to the investigators.

Adverse events

There were no serious adverse events and no adverse events that could be related to use of the sprays. Several parents reported dry flaky skin on their children’s scalps. This was not considered treatment related as screening in school showed that most children had flaky scalps and the parents only noticed this while combing to check for lice. One child experienced an unexpected rash on her neck but her mother did not think it was treatment related, stopped spraying for a few days, and then continued with no further incidence. An outbreak of chicken pox occurred in the community during late June and early July 1994, which caused some parents to stop spraying on a temporary basis.

Discussion

We have conducted two field studies evaluating a spray designed to repel head lice. Our proof of concept suggested that 2% piperonal might reduce the incidence of reinfestation for short periods but the study could not run for long enough to properly evaluate its effectiveness in a population with a high prevalence of head louse cases. The double-blind randomised study, with a moderately high reinfestation risk, suggested that regular use of the product may have offered some benefit, although the differences between the groups were not significant (p < 0.05) at any level. Our knowledge of the families suggested that any observed benefit was probably mostly related to the diligence of the carer in using the product.

Despite most parents wanting a product that helps prevent infestation, no louse repellent product had been developed previously (Peock & Maunder, 1993; Canyon & Speare, 2007). Because nobody knows when children are at risk of infestation it would need to be used more or less continuously when children have contact with others. Increased risk occurs occasionally such as “outbreaks” in schools and when attending parties and sleepovers, meeting new contacts and for children spending more time in close proximity with their peers than normal (Parison, Speare & Canyon, 2013). We found that many parents are more concerned about the risk of lice from school rather than social contacts so they were less likely to apply repellent before the children went to parties and at weekends when children regularly visited their friends.

Piperonal is a novel, pharmacologically safe repellent, widely used as a fragrance and flavouring agent for cosmetics and foodstuffs, with an acceptable odour. Piperonal melts at 35 °C–39 °C, so when sprayed on hair it is borderline to melting, needing formulation to maintain a fluid state. It is physically and chemically stable and slightly more volatile than the flying insect repellent N, N-diethyl-3-methylbenzamide (DEET) but, unlike DEET, piperonal is not absorbed transdermally. During the investigations conducted as part of the development of the 2% piperonal product it was compared in vitro with other putative repellents. We tested several apparently identical samples of DEET for repellence against lice but unlike the observations of Canyon & Speare (2007) they were found to exhibit variable levels of activity ranging from similar to piperonal to no effect at all, with the majority in the latter category (IF Burgess, 1993, unpublished data). The study sponsor checked each of these samples by GCMS for purity and chemical consistency and no differences were detectable by this method. There is also one enigmatic report suggesting that 3-(N-acetyl-N-butyl)aminopropionic acid ethyl ester (IR3535) may be more repellent to lice, but the article omits relevant data (Bohlmann, 2008).

Both our studies involved communities where the prevalence of infestation was high, effectively 100% in Dhaka and around 22%–25% during 1993 in the North London index school. However, at a school examination near the end of the study (October 1994), when we expected high infestation following summer holiday family visits, the prevalence was just 10.6%, suggesting that increased vigilance by parents during the study to deal with any cases of infestation quickly had reduced transmission enough to impact on overall prevalence. It is unlikely that the small observed repellent effect had played a role in this reduction.

For most potential repellent users, the underlying risk of infestation is lower than in our investigated communities, as European surveys have indicated (Smith et al., 2003; Harris, Crawshaw & Millership, 2003; Buczek et al., 2004; Willems et al., 2005; Jahnke, Bauer & Feldmeier, 2008; Rukke et al., 2011). Consequently, for consumer satisfaction a repellent could be less effective like, for example, a mosquito repellent. But, are repellents worth the cost and time involved in correct and thorough application, plus continued vigilance to confirm their effectiveness? Perhaps just checking the children’s hair regularly and treating any lice found would be better?

Dethier defined repellence as “.. any stimulus that elicits an avoiding reaction may be termed a repellent” (Dethier, 1947). This includes physical and chemical effects but recent public interest in use of natural and plant extracts has resulted in targeting essential oils as repellents, mostly based on folklore and ancient herbals. There is no scientific basis for this because volatile oils from plants are believed to have evolved as feeding deterrents to phytophagous insects or as attractants for pollinators (Dethier, 1947). Consequently, the idea they would repel haematophagous insects is speculative. Volatile oils confuse host seeking flying insects but crawling obligate ectoparasites like lice do not “host seek”. Their migrations are triggered by physical stimuli such as movements of the hair signalling contact of one host with another (Szczesna, 1978; Burgess, 1995). Lice may not detect odours from a potential new host so they would play no role in the transfer process, meaning chemical deterrents must exert potent effects on the sensory physiology of lice to stop them moving onto treated hair. Physically repulsive materials, e.g. heavy oils, may be more deterrent than volatile materials, and some volatile materials may just be chemical irritants rather than true repellents (Canyon & Speare, 2007; Canyon, 2010), although antennectomy indicated that odour plays some role in the louse’s response to chemicals like piperonal (Peock & Maunder, 1993).

Pre-clinical tests of repellents have difficulty mimicking the natural substrate of hair on a head and lice become acclimated to an odour. An effective repellent must deter head lice within seconds, or at most minutes, of first contact. Therefore, laboratory tests lasting several hours are irrelevant to practical deterrence of head lice, although they could be applied to deterring body/clothing lice (Semmler et al., 2010; Semmler et al., 2012). Generally essential oils, single terpenoids, and aliphatic lactones exhibit no more repellence in vitro than piperonal (Toloza et al., 2006a; Toloza et al., 2006b; Toloza et al., 2008). However, most of these compounds are also insecticidal (Canyon & Speare, 2007; Canyon, 2010; Semmler et al., 2010; Semmler et al., 2012; Toloza et al., 2006a; Toloza et al., 2006b; Toloza et al., 2008; Mumcuoglu et al., 1996), so some reported repellence may be a misinterpretation of toxicity, e.g. a field study using 3.7% citronella, a concentration that is often insecticidal, may actually have only recorded insecticidal activity against invading lice (Mumcuoglu et al., 2004).

It should be remembered that our first observations were made while investigating the pediculicidal effects of piperonal that had been previously reported decades earlier (Corlette, 1925; Burgess, 1993a). So, if study participants had applied piperonal spray more thoroughly, would the outcome have been improved through accidentally killing lice rather than repelling them? We shall never know. However, as manufacturers and consumers continue to hope for a new preventive product, a piperonal-based spray repellent has recently been launched in Australia (Pharmacare Laboratories Pty Ltd, 2012), which in view of our experience is unfortunately unlikely to prove more effective than the product we tested, unless used rather more thoroughly than we observed.

Supplemental Information

Supplemental Information 1 Click here for additional data file.

Supplemental Information 2 Protocol for the London study

Click here for additional data file.

Supplemental Information 3 CONSORT checklist

Click here for additional data file.

Investigation team members who contributed to the studies but are not named as authors include Ayesha Akhter Ruma, Nasrine Khan, Jafour Iqbal Khan, and the late Dr Nur Islam who was chair of the ethics committee and also acted as medical supervisor (Dhaka); Barbara Shenkin, Susan Peock, and Dr JS Adler who acted as medical supervisor (North London). We wish to thank the various organisations that hosted or facilitated the work in Dhaka including Bottomley Home, Farmgate; Ardasha Islami Mission, Mohammedpur; Gandaria community; and German Doctors for Developing Countries. Also our thanks go to Beis Yaakov Primary School and other schools and institutions in the Golders Green, Colindale, and Edgware areas of North London. The decision to publish the study was that of the authors, with no input from the original sponsor or any of its ex-employees into the content of the writing or the new analyses. Any opinions expressed are those of the authors.

Additional Information and Declarations

Competing Interests

Author Contributions

Clinical Trial Ethics

Patent Disclosures

Ian F. Burgess, Nazma A. Burgess and Christine M. Brown are or were employees of Medical Entomology Centre, Insect Research & Development Limited; and Judith Kaufman is an employee of Royalheath Charitable Trust Limited.

Ian F. Burgess conceived and designed the experiments, performed the experiments, analyzed the data, wrote the paper, prepared figures and/or tables, reviewed drafts of the paper.

Christine M. Brown conceived and designed the experiments, performed the experiments, reviewed drafts of the paper.

Nazma A. Burgess conceived and designed the experiments, performed the experiments, reviewed drafts of the paper, provided community liaison Bangladesh, translation services.

Judith Kaufman conceived and designed the experiments, performed the experiments, reviewed drafts of the paper, provided community liaison London, pharmacy services.

The following information was supplied relating to ethical approvals (i.e., approving body and any reference numbers):

Ethical approval in Dhaka was granted by the ad hoc ethics committee of the Metropolitan Medical Centre, Mohakhali, Dhaka; Protocol RAP001.

Ethics approval in North London was granted by Barnet Research Ethics Committee; Protocol RAP002.

The following patent dependencies were disclosed by the authors:

Irwin RN. 1992. Benzodioxolane pesticides and pest repellents. UK Patent, GB 2 270 843.

Irwin RN. 1993. Louse repellent compositions. International Patent, WO 9502960 A1.

Oliver WJ. 1992. Louse repellent composition. UK Patent, GB 2 267 643 A.

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
