# Peer review of "Can head louse repellents really work? Field studies of piperonal 2% spray"

_PeerJ, doi:10.7717/peerj.351_

## Round 0.1 · original submission · Major Revisions

A range of major issues need to be resolved before the manuscript is ready for publication.

In the statistical analysis section Line 151. This is the first time we hear of the sponsor BIOS – and it seems that they have all the data? It is not clear as to what data was collected by the authors and what was collected by BIOS – this needs to be clarified. I am very concerned that the data you are showing here is not raw data. Can the data that was used be made accessible – ie on Figshare? Your statistical methods are generally quite unusual – can you provide references for your statistical methods please.
Other comments l ine 87/88 reference for the OECD guidelines needed

Line 108 – Objectives should be in the introduction
Line 167 – 203. This is all methodology. Not results.
Line 206 – ‘surprisingly low’. Is this result statistically significant or just an observation? There is no data to back this statement up –
Line 215. There is no mention in the methods of the use of a chi-sq test. Your results need to reflect the methods you used. This is also relevant for other parts of your results
Line 231 – 235 – Methods
Line 237 - a ‘slight non-significant trend’ with a p-value of 0.4368! That is not correct!
Line 240 – ‘ marginally increased difference’ – with a p-value of 0.3323. Again are you really sure what you are saying here?
Adverse events section – again this is not results – this is methods
Line 265 – 272. There is no data of references to back up this point. It is all descriptive notetaking.

re: clinical trial or not - really from the methods it is very hard to tell what data was actually collected. Clearer methods are required to make a determination on this.

Please address Reviewer 2's comments.

Reviewer 1 ·

Basic reporting

No comments.

Experimental design

No comments.

Validity of the findings

No comments.

Additional comments

The paper can be accepted as it is.

Reviewer 2 ·

Basic reporting

Overall: Reinfection is thought to be a significant pediculosis issue and so it was nice to see some old data on the topic being submitted to add more evidence to the pool.

Experimental design

Due to the lack of empirical data on reinfection, this paper contains results worthy of publication despite there being design flaws, such as the lack of demographics associated with the data and lack of consistently applied methodology between studies and between study participants. Other limitations that should be identified include the lice detection and elimination methods which are known not to be accurate.

Validity of the findings

Introduction
P1: Another reason is that it is easy to miss a recent infection because it is hard to detect lice and no few eggs are present.
P2: It has already been shown that heavy oils are not a deterrent to head lice transmission – see Canyon et al. Int J Dermatol 2007;46:422-426. You actually refer to this in Discussion P6.

Methods
P2: What date were data collected?
P2: How was the infection rate of close to 100% ascertained?
P3: What date were data collected?
P3: Clarify which species was involved with the resistance studies.
P3: Why was an average infection rate provided for only “some” schools and not all schools assessed?
P3: What was the nature of the information on reinfection and from how many sources (parents?) did it originate?
P4: Given that plastic comb is not a great way to detect an infection, the number of lice per head on participants must have been considerably higher than average – true?
P13: The dose must have varied considerably due to 5-25 sprays being administered per head – comment?
P14: Provide reference for the study finding that vanillin was not repellent.
P15: Provide a reference to show that the use of carbaril 1% actually eliminates 100% of all lice. How was the elimination of lice confirmed? The only reliable method is combing with conditioner to see if eggs or lice remain.
P18: The paragraph on objectives does not belong in the methods section and should be moved to the end of the introduction.
P20: Since the dates of data collection were not supplied it is not possible to determine the accuracy of the statement that “No studies had ever been conducted of incidence of head lice infestation…”.
P21: Again, how was the 100% infection rate confirmed or estimated?
P21: Recruitment does not equate to reinfection risk – rewrite.
P21: Provide numbers of study participants.
P22: To what groups are you referring?
P28-29: Presumably you are talking about the Bangladesh data in this and the next paragraphs?

Results
P1: The selection of children did not sound random – comment?
P12: The dose is critical so this lack of design integrity is a major flaw in the study design – there is no way to know if reinfection occurred on heads that were sprayed more or less.

Discussion
P1: We know that reentry to school after a break is correlated with a higher occurrence of pediculosis – see publications by Speare et al on school programs.
P3: DEET’s low efficacy against lice – cite published data when it exists not unpublished data (Canyon et al. Int J Dermatol 2007;46:422-426)
P4: 10% prevalence is a global average (see Gratz) which is not low, but expected.
P4: Disagree – since the bulk of infections involve up to ten lice per head, a repellent is completely useless if it is not 100% effective. Even the passage of a single fertilized female would rapidly result in a new outbreak on a head. Consumer satisfaction is irrelevant because they often even fail to detect infections.
P5: First sentence is not accurate – the global infection rate is around 10% in most schools rising to much higher rates and lower rates in pockets.
P6: Refer to more up to date descriptions of repellence in Canyon et al. Int J Dermatol 2007;46:422-426
P7: See Canyon et al in J Investigative Dermatol 2002; 119(3): 629-631 for a more evidence-based statement on the speed of transmission

Additional comments

Writing: Grammar requires improvement throughout. While I used to use the words “infested” and “reinfestation”, I have come to see that they have negative connotations and that “infection” and “reinfection” are preferable since we are talking about pediculosis which is a disease.

---

## Round 0.2 · accepted · Accept

I am happy with your responses to the issues raised, and it can now head out for publicaiton